# Antioxidant Activity in Extracts from *Zingiberaceae* Family: Cardamom, Turmeric, and Ginger

**DOI:** 10.3390/molecules28104024

**Published:** 2023-05-11

**Authors:** Pura Ballester, Begoña Cerdá, Raúl Arcusa, Ana María García-Muñoz, Javier Marhuenda, Pilar Zafrilla

**Affiliations:** Faculty of Pharmacy and Nutrition, Universidad Católica San Antonio de Murcia (UCAM), Campus de los Jerónimos, Guadalupe, 30107 Murcia, Spain; pballester@ucam.edu (P.B.); bcerda@ucam.edu (B.C.); amgarcia13@ucam.edu (A.M.G.-M.); jmarhuenda@ucam.edu (J.M.); mpzafrilla@ucam.edu (P.Z.)

**Keywords:** antioxidant activity, *Elettaria cardamomum* L. Maton (cardamom), *Curcuma longa* L. (turmeric), *Zingiber officinale* Roscoe (ginger), bioactive compounds

## Abstract

An increase in life expectancy leads to a greater impact of chronic non-communicable diseases. This is even more remarkable in elder populations, to whom these become main determinants of health status, affecting mental and physical health, quality of life, and autonomy. Disease appearance is closely related to the levels of cellular oxidation, pointing out the importance of including foods in one’s diet that can prevent oxidative stress. Previous studies and clinical data suggest that some plant-based products can slow and reduce the cellular degradation associated with aging and age-related diseases. Many plants from one family present several applications that range from the food to the pharmaceutical industry due to their characteristic flavor and scents. The *Zingiberaceae* family, which includes cardamom, turmeric, and ginger, has bioactive compounds with antioxidant activities. They also have anti-inflammatory, antimicrobial, anticancer, and antiemetic activities and properties that help prevent cardiovascular and neurodegenerative diseases. These products are abundant sources of chemical substances, such as alkaloids, carbohydrates, proteins, phenolic acids, flavonoids, and diarylheptanoids. The main bioactive compounds found in this family (cardamom, turmeric, and ginger) are 1,8-cineole, α-terpinyl acetate, β-turmerone, and α-zingiberene. The present review gathers evidence surrounding the effects of dietary intake of extracts of the *Zingiberaceae* family and their underlying mechanisms of action. These extracts could be an adjuvant treatment for oxidative-stress-related pathologies. However, the bioavailability of these compounds needs to be optimized, and further research is needed to determine appropriate concentrations and their antioxidant effects in the body.

## 1. Introduction

The use of natural bioactive compounds to prevent and improve health is widely accepted. Traditional medicine has always found in nature an important source of remedies for numerous pathologies. This is the basis for the use of nutraceuticals as adjuvants in these conditions. The *Zingiberaceae* family, comprising more than 1400 species distributed in tropical and subtropical areas, has been widely used in traditional Chinese and Indian medicine to prevent and treat numerous pathologies [1,2]. Plants of this family are also used as foods (spices and herbs) and to produce natural dyes [3].

To this family belongs cardamom, turmeric, and ginger. These plants are among the most active natural remedies due to their numerous biological properties. The following activities are attributed to them: antioxidant, anti-inflammatory, anticancer, antigrowth, antiarthritic, antiatherosclerotic, antidepressant, antiaging, antidiabetic, antimicrobial, wound healing, and memory enhancing [4,5].

The most used varieties of cardamom are green and black. The main bioactive compounds of cardamom are terpenes and phenolic compounds, which exhibit powerful antioxidant properties. Their main bioactive compounds are as follows: 1,8-cineole, α-terpinyl acetate, nerolidol, sabinene, g-terpinene, α-pinene, methyl linoleate, α-terpineol, β-pinene, *n*-hexadecanoic acid, and limonene [6]. 1,8-cineole and α-terpinyl acetate are the principle bioactive components in black and green cardamom (Table 1). These compounds aid in safeguarding against chronic diseases and oxidative stress (OS) [6,7]. Additionally, cardamom has been found to improve insulin sensitivity and glucose uptake [8], which may help to reduce the risk of type 2 diabetes [9].

Turmeric has a very complex and diverse composition; principally containing terpenoids and phenolic compounds, it also contains other compounds, such as carbohydrates (sugars and fiber), proteins, minerals, and resin [10]. Mainly, turmeric bioactive compounds are curcuminoids (curcumin I, II, and III) and non-curcuminoid compounds [11] (Table 1). Curcumin has been shown to possess strong antioxidant activities that help protect against OS and various chronic diseases [12,13,14]. Curcumin is a powerful scavenger of free radicals (FR), which are unstable molecules that damage cells and contribute to the development of chronic diseases such as cancer, heart disease, and diabetes [15,16,17]. Non-curcuminoid compounds [11] also present significant biological activity and have been attributed with anti-inflammatory, antioxidant, and anticancer properties [10].

The main bioactive compounds of ginger are phenolic compounds and terpenes (Table 1). In addition to these it contains polysaccharides, such as fiber, lipids, and organic acids [18]. Its most bioactive phenolic compounds are gingerols, shogaols, and paradols. The first ones are found in ginger and from them the others are obtained by heat treatment and hydrogenation, respectively. Many beneficial properties have been attributed to ginger, including treatments for arthritis, muscular aches, pain, sore throat, cramps, hypertension, dementia, fever, infectious diseases, cough, nervous diseases, gingivitis, asthma, and gastric disorders, among others [19,20,21]. Both gingerols and shogaols exhibit a host of biological activities, ranging from anticancer, antioxidant, antimicrobial, anti-inflammatory, and anti-allergic to various central nervous system activities [22].

**Table 1 molecules-28-04024-t001:** Bioactive compounds of cardamom, turmeric, and ginger adapted from [6,10,23].

Plants	Dose	Phytochemicals	Bioactive Compounds
Cardamom	1–3 g/d	Terpenes	1,8-cineole, α-terpinyl acetate, nerolidol, sabinene, *g*-terpinene, α-pinene, methyl linoleate, α-terpineol, β-pinene, *n*-hexadecanoic acid, and limonene
		Phenolic compounds	Curcumin I (1,7-bis(4-hydroxy-3-methoxyphenyl)-1,6-heptadiene-3,5-dione)
Turmeric	0.5–2 g/d	Curcuminoid (polyphenolic compounds)	Curcumin II (demethoxycurcumin, 1-(4-hydroxy-3-methoxyphenyl)-7-(4-hydroxyphenyl)-1,6-heptadiene-3,5-dione, and curcumin III (bisdemethoxycurcumin, 1,7-bis(4-hydroxyphenyl)-1,6-heptadiene-3,5-dione)
		Non-curcuminoids (terpenoids and phenolic compounds)	Turmerones, elemene, bisacurone, curdione, cyclocurcumin, germacrone, furanodiene, curcumol, and calebin A
Ginger	1–3 g/d	Phenolics compounds	Gingerols, shogaols, paradols, quercetin, zingerone, and gingerenone-A
		Terpene components	β-bisabolene, α-curcumene, zingiberene, α-farnesene, β-sesquiphellandrene, and zingiberene

Overall, the *Zingiberaceae* family offers a wealth of health benefits, which turns into a valuable addition to a healthy diet, as its beneficial effects are enhanced by its culinary use as a spice, making it a tasty and convenient way to add antioxidants to one’s diet [24].

Regarding the bioavailability of these compounds, curcumin has a poor pharmacokinetic profile and low bioavailability in humans, even at high doses. It is chemically unstable, poorly soluble in water, and degrades at an alkaline pH. Therefore, curcumin is poorly absorbed in the small intestine, combined with a high hepatic reductive and conjugative metabolism that clears curcumin components quickly by biliary and urine excretion [25]. Only a small amount of curcumin is distributed from blood to tissue [26]. Several authors [27] have observed that a dose of 8 g of turmeric has no adverse effects in humans.

The bioavailability of curcumin depends on the types of molecular complexes it creates. It forms them with carbohydrates, proteins (such as whey proteins, bovine serum albumin, and β-lactoglobulin), lipids (phospholipids), and bioactive compounds, such as resveratrol, quercetin, and piperine, and the formation of these contributes to increasing the bioavailability of curcumin [27].

The absorption of curcumin is multiplied by 10 times when ingested together with piperine [28]. Piperine is the alkaloid responsible for the pungency of black pepper that inhibits glucuronidation processes by the enzyme UDP-glucuronosyltransferase in the small intestine and liver [12].

The amount of curcumin distributed from the blood to the tissues is very small, below the detection limit. Liposomes, capable of carrying lipophobic and hydrophobic particles, are being used to improve the bioavailability of curcumin. Curcumin analogues can also be used, modifying their chemical structure to improve absorption. Curcumin analogues include the following: bis-o-hydroxycinnamoylmethane, bis-1.7-(2-hydroxyphenyl)-hepta-1.6-diene-3.5-dione (BDMC), tetrahydrocurcumin (THC) analogs, and two new curcumin derivatives, B06 and C66, which have been shown in studies to have higher bioavailability [29].

Gingerols (the main compounds of ginger) have the problem of low solubility, especially 6-gingerol. It is important to optimize them through different techniques, such as nanotechnology, to increase their bioavailability [30,31,32].

With respect to the metabolization of the main compounds of ginger root (6-, 8-, and 10-gingerol and 6-shogaol), it is observed that glucuronide-conjugated metabolites are the major metabolites, and sulfate-conjugated metabolites are also detected [17]. These authors have observed that these metabolites are detected in blood samples after 15 min of consumption. These authors did not detect 6-shogaol or free 6-8-10 gingerols.

Several authors have observed that the main metabolic pathway of shogaols and gingerols is ketone reduction, with gingerdiols being the main metabolites. Conjugation with cysteine is one of the main metabolic pathways of 6-shogaol (90%). We could consider these metabolites as responsible for the beneficial effects of gingerols and shogaols [25].

Zhang et al. 2022 have observed in asthmatic patients that 33% of 6-gingerol is metabolized to its reducing metabolites, which are the 6-girgerdiols. 6-shogaol is metabolized to its phase I metabolites and conjugated to cysteine. The half-life of these compounds ranges from 0.6 to 2.4 h [26]. Chen et al. [33] showed that hydrogenation, oxidation, and demethylation metabolisms of gingerols are the main metabolic types in microsomes. However, there are differences in metabolic kinetics and the metabolic types of different species of liver microsomes. CYP2C19 and CYP1A2 are the main enzymes involved in the oxidation and demethylation metabolism of these gingerols, but the affinity of the gingerols is not balanced.

Bioactive compounds in cardamom also have low bioavailability due to complex molecular structure [34]. Their pharmaceutical and food applications are limited by the low solubility of their active components in aqueous media. Encapsulation could be a solution to improve their bioavailability and stability [6].

The objective of this review is to establish and unify in a single comprehensive review how the antioxidant capacity of the *Zingiberaceae* family, particularly turmeric, ginger, and cardamom, could contribute to health-associated benefits by the intake of their bioactive compounds through diet.

## 2. Relationship between Oxidative Stress and Health

OS is known as an imbalance between oxidizing agents and antioxidants in favor of the first [35]. This imbalance may be due to a low intake of antioxidants, depletion of endogenous antioxidants, or an increase in reactive species (RS) [36]. Consequently, it can induce damage to lipids, proteins, and DNA [37].

Among RS [38] are highlighted those of oxygen (ROS) [39], synthesized in cells during mitochondrial oxidative metabolism, and those of nitrogen (RNS), synthesized under hypoxia conditions [40]; there are also RS of sulfur (RSS) [41], of carbonyl (RCS) [42], and of selenium (RSeS) [43]. It is known that under physiological conditions, the organism can synthesize ROS in small quantities, which is afterwards neutralized. ROS are involved in vital processes for the organism, such as cellular homeostasis, gene expression, receptor activation, or signal transduction. Excessive and prolonged production of ROS and RNS may turn into damage to cell structure and function, which can lead to irreparable changes. For instance, lipid peroxidation of membrane lipids can impair membrane function, inactivate membrane receptors and enzymes, increase permeability to ions, decrease membrane fluidity, and ultimately rupture the membrane [39].

There are different types of SRs, and whenever possible it is convenient to use the name of the specific chemical species instead of using the term FR, since depending on their origin, some are radical (they have an unpaired electron) and others are non-radical (they are the product of the reduction of two electrons), as shown in Table 2 [38].

**Table 2 molecules-28-04024-t002:** Classification of the most common reactive species. Adapted from Sies and collaborators [38].

Free Radicals	No Radicals
Reactive Oxygen Species	
Superoxide anion radical (O_2_¯˙)	Hydrogen peroxide (H_2_O_2_)
Hydroxyl radical (OH˙)	Singlet molecular oxygen (O_2_^1^ ∆_g_)
Peroxyl radical (ROO˙)	
Reactive Nitrogen Species	
Nitric oxide (NO˙)	Nitrite (NO_2_¯)
Nitrogen dioxide (NO_2_˙)	Peroxynitrite (ONOO¯)

FRs are highly unstable molecules due to the presence of one or more unpaired electrons, and as we have seen they can come from oxygen and nitrogen, among other elements [44]. They have a short lifespan, ranging from milliseconds to nanoseconds, and in order to stabilize they seek coupling with electrons from contiguous biological molecules, with donating or accepting electrons being the genesis of OS [45].

When the production of ROS is excessively high and the organism is not able to neutralize them, harmful effects begin to appear in vital cellular structures such as proteins, lipids, and nucleic acids [46], producing OS. OS is responsible for the appearance and progression of different types of diseases, such as atherosclerosis, cardiovascular diseases, diabetes, cancer, and metabolic disorders [47].

ROS production originates from enzymatic and non-enzymatic reactions. Once O_2_¯˙ is synthesized, it is involved in certain reactions that produce radical and non-radical RS, such as the following: H_2_O_2_, OH˙, ONOO¯, etc. OH˙ is the most reactive among all FR species in vivo. FRs are synthesized from endogenous and exogenous sources and their sources are summarized in Table 3 [48].

**Table 3 molecules-28-04024-t003:** Endogenous and exogenous sources that generate FRs [48].

Endogenous Sources	Exogenous Sources
Inflammation	Exposure to contaminants
Ischemia	Exposure to heavy metals (Hg, Cd, Pb, Fe, and As)
Infection	Chemical solvents
Cancer	Food cooking (smoked meat and used oil)
Activation of immune cells	Medications (cyclosporine, tacrolimus, bleomycin, and gentamicin)
Excessive exercise	Cigarette smoke
Mental stress	Alcohol
Aging	Radiation

Hg: mercury; Cd: cadmium; Pb: lead; Fe: iron; As: arsenic.

As mentioned above, an uncontrolled excess of RL will produce OS. For example, an excess of OH˙ and ONOO¯ could cause lipid peroxidation, damaging cell membranes and lipoproteins and creating malondialdehyde (MDA) and conjugated diene compounds, which are cytotoxic and mutagenic [48]. As a radical chain reaction, lipid peroxidation propagates extremely rapidly, affecting a large number of lipid molecules [49]. OS can also cause DNA damage through the formation of 8-oxo-2′-deoxyguanosine (8-OHdG), responsible for mutagenesis [50] as well as loss of epigenetic information [51]. Oxidative DNA damage is also a stimulus in the development of cancer [52]. OS is also considered a primary or secondary cause of a large number of CVDs [53], as circulating LDL oxidized by ROS contributes to foam cell formation and lipid accumulation, resulting in the formation of atheromatous plaques. OS has also been associated with various neurological diseases, such as Alzheimer’s disease (AD), Parkinson’s disease, amyotrophic lateral sclerosis, multiple sclerosis, depression, and memory loss [54]. OS is implicated in renal diseases, such as renal failure, proteinuria, and uremia [55,56], as well as respiratory diseases, such as asthma and chronic obstructive disease [57], and delayed sexual maturation and onset of puberty [58]. Thus, the excess of FRs and consequently OS affect different tissues, being responsible for the onset and progression of different pathological conditions that affect human health.

Several biomarkers have been employed and exhaustively examined to evaluate OS in reference to different diseases [35,59], among the most important of which are carbonyl groups and xanthine oxidase (XO) in proteins, advanced glycation end products (AGEs) in carbohydrates, 8-OHdG in DNA and MDA, and oxidized low-density protein (OxLDL) and F2-isoprotans in lipids.

Antioxidants are used to combat oxidation. Antioxidants are stable molecules that have the capacity to donate an electron to an FR in order to neutralize it, reducing its potential damage [60]. Antioxidant systems can be produced endogenously in the organism or introduced exogenously [61]. Regarding the endogenous form, the organism possesses mechanisms of action to neutralize RS through enzymatic and non-enzymatic mechanisms [44]. Among the enzymatic mechanisms, referred to as high-molecular-weight mechanisms, are as follows: superoxide dismutase (SOD), the main antioxidant enzyme of the organism, catalyzes the superoxide anion into oxygen (O_2_) and H_2_O_2_ and in turn interacts with different enzymes that help neutralize SRs; catalase (CAT), which catalyzes H_2_O_2_ in water and O_2_ in the presence of nicotinamide adenine dinucleotide phosphate (NADPH) oxidase; glutathione peroxidase (GPx), which reduces H_2_O_2_ and organic peroxides in water or alcohol whenever selenium is present; and peroxiredoxin (Prx), which shows the lowest efficiency in H_2_O_2_ metabolization [44,60,62]. As for low-molecular-weight or non-enzymatic mechanisms, they include vitamin E, vitamin C, bilirubin, biliverdin, uric acid, ascorbic acid, glutathione, and flavonoids [44,63].

## 3. Antioxidant Activity of *Zingiberaceae* Family

### 3.1. Antioxidant Activity of Ginger

Ginger has been extensively studied for its antioxidant properties. The underlying mechanism of action lies in the fact that ginger possesses bioactive compounds with two or more hydroxyl groups attached to an aromatic ring that confer on them the antioxidant capacity to neutralize unpaired electrons of FRs and donate hydrogens or chelate metal ions, thus reducing OS [18]. If not neutralized, these FRs, such as ROS or RNS, interact with cells, causing oxidative damage, which can contribute to cellular aging and the development of chronic diseases, such as cancer [64], inflammatory diseases [65], neurodegenerative diseases [66], and cardiovascular diseases [67]. This antioxidant property of ginger is due to the presence of antioxidant compounds, such as gingerols (6-gingerol, 8-gingerol, and 10-gingerol), shogaols (6-shogaol, 8-shogaol, and 10-shogaol), flavonoids, and phenolic acids [68].

As previously mentioned, oxidative damage can affect proteins and cell membrane lipids [69]. Proteins can be oxidized by the formation of FR groups in their component amino acids, which can alter their structure and function; this can also damage the structure of the cell membrane and increase its permeability. It has been observed that ginger may have a protective effect against this oxidation in several studies. This has been investigated in the study conducted by Romero et al., where it was observed that a ginger extract decreased AKT-phosphorylation, thus protecting cells from hydrogen peroxide-induced oxidative damage in human hepatocyte cell cultures [70]. In another study carried out by Carnuta et al., it was shown that a *Zingiber officinale* extract reduced lipid accumulation in hamsters’ livers, caused by a decrease in endoplasmic reticulum stress (ERS) markers [71].

Cells employ antioxidant mechanisms, such as MDA, or enzymes, such as SOD, CAT, and GPx, to compensate for oxidative damage. These antioxidants can neutralize FRs and reduce OS in cells. For example, a meta-analysis of clinical trials has shown that ginger supplementation decreases MDA values and increases total antioxidant capacity [72]. This is consistent with the findings of the study conducted by Sheikhhossein et al., in which there was also a significant reduction in MDA; in addition, an increase in GPx was observed [73].

In addition to ROS, reactive nitrogen species, including nitric oxide (NO), are known to contribute to the development of diseases by influencing signal transduction and causing DNA damage. The production of NO is regulated by inducible nitric oxide synthase (iNOS), which is upregulated in response to various stressors [74]. Oxidative damage to DNA can lead to mutations and activation of cell signaling pathways that may contribute to the development of chronic diseases, such as cancer, because FRs can cause strand breaks and oxidation of DNA’s nitrogenous bases, which can lead to errors in replication and the formation of mutations. Studies have demonstrated that 6-gingerol can effectively inhibit XO, which is responsible for catalyzing the oxidation of hypoxanthine to xanthine and then xanthine to uric acid. This process occurs during the final stage of purine metabolic degradation and results in the production of ROS [21].

In addition, other studies have shown that 6-gingerol inhibits NO production and reduces iNOS expression in lipopolysaccharide (LPS)-stimulated mouse macrophages [75]. 6-gingerol was found to effectively suppress oxidative damage mediated by peroxynitrite. Other ginger derivatives, such as 6-shogaol, 1-dehydro-10-gingerdione, and 10-gingerdione, were also found to decrease LPS-induced NO production, with 6-shogaol and 1-dehydro-10-gingerdione effectively reducing iNOS expression [76].

During an inflammatory response, leukocytes and macrophages release ROS to destroy invading microorganisms. However, ROS can also damage surrounding cells and tissues, which can lead to a chronic inflammatory response and contribute to the development of chronic inflammatory diseases, such as arthritis, inflammatory bowel disease, or systemic lupus erythematosus, among others [77].

Chronic inflammation causes the elevation of pro-inflammatory markers, such as nuclear factor-kappaB (NF-κB), the mechanistic target of rapamycin (mTOR) pathways, interleukin (IL)-6, total antioxidant capacity (TAC), c-reactive protein (CRP), and tumor necrosis factor alpha (TNF-α). Several studies have shown that ginger can reduce the concentrations of these markers, and this has been supported by several studies, such as the systematic review and meta-analysis carried out by Jalali et al. [72]. In another study, it was demonstrated that 6-gingerol enhances the expression of Beclin1, which promotes autophagy in human endothelial cells. It also hinders signaling in the PI3K/AKT/mTOR pathway without affecting the cell cycle [78]. In addition, it has been shown that ginger extract can inhibit NF-κB activation and decrease the level of IL-1β [79]. The anti-inflammatory effects of 6-shogaol were achieved through the inhibition of the production of PGE2 and pro-inflammatory cytokines (IL-1β, IL-6, TNF-α, and cyclooxygenase-2 (COX-2)). This was accomplished by downregulating the expression of p38MAPK and NF-κB [80].

The anticancer properties of phenolic compounds present in ginger have also been demonstrated. Some studies have shown that 6-gingerol and 6-shogaol produce cell cycle arrest, inducing growth suppression and a decrease in tumor volume and tumor burden through the increase in CKDIs p21 (Cyclin-Dependent Kinase Inhibitors), which are involved in the control of the transition of cells from the G1 to the S phase of the cell cycle [81,82,83]. In addition, the benefits of ginger as an antiemetic have been described in several controlled studies. A systematic review and meta-analysis realized by Kim at al. showed that ginger or its bioactive compounds diminish acute and delayed phases of chemotherapy-induced nausea and vomiting nausea in breast cancer patients [84], producing inhibition of 5-hydroxytryptamine type3 (5-HT3) receptors in the central and peripheral nervous systems. The efficacy of ginger on postoperative nausea and vomiting [85] and on nausea and vomiting of pregnancy [86] has also been demonstrated.

Inflammation and ROS are closely related to the pathogenesis of cardiovascular disease (CVD). Chronic inflammation can trigger the production of ROS, activate several inflammatory signaling pathways, and promote the expression of pro-inflammatory molecules. This is why the use of ginger has been extensively studied in this type of disease [87]. Studies have shown that several bioactive compounds found in ginger exhibit hypolipidemic, anti-inflammatory, and antioxidant effects in vivo. For example, El-Seweidy et al. conducted a study that revealed that 10-dehydrogingerdione possesses a proprotein convertase subtilisin/kexin type 9 (PCSK9) lowering effect, which may be responsible for its cardiovascular protective effect against dyslipidemia through a mechanism that modulates lipid levels [88].

Neurodegenerative diseases are also related to increased ROS production in the brain. This increase can cause damage to cells and tissues, which has been linked to brain cell death and neuronal dysfunction in neurodegenerative diseases [66].

Ginger has been studied in these types of diseases. It has been observed that 6-shogaol inhibited some components of the inflammatory pathway, such as TNF-α, NO, COX-2, and iNOS, in patients with Parkinson’s disease [89]. In addition, it has been suggested that this compound may inhibit glial cell activation and reduce memory impairment in animal models of dementia [90], as well as exert a protective effect against amyloid-β in AD [91].

The antimicrobial properties of ginger have been well documented in recent years, with various studies highlighting its effectiveness against a range of pathogens [92]. Bioactive compounds present in ginger are believed to target the cell membrane and cell wall of microorganisms, leading to the disruption of their structural integrity and eventual cell death [93]. Ginger has demonstrated broad-spectrum antimicrobial activity against Gram-positive and Gram-negative bacteria, as well as fungi and viruses [92]. This can be attributed to its ability to inhibit bacterial protein synthesis, DNA replication, and cell division [94]. Furthermore, ginger has been found to be synergistic with conventional antibiotics in combating resistant strains of bacteria [95].

In addition to its direct antimicrobial effects, ginger also possesses anti-inflammatory and immunomodulatory properties, which can boost the host’s immune response against infections [94]. This makes ginger a promising candidate for the development of new antimicrobial agents, particularly in the face of increasing antibiotic resistance [92].

Table 4 and Figure 1 summarize the properties and mechanisms of action of ginger.

**Table 4 molecules-28-04024-t004:** Summary of product characteristics of ginger.

Extract	Function	Mechanism	References
Ginger	Antioxidant	Decreased AKT-phosphorylation	[70]
		Decreased ERS markers	[71]
		Decreased MDA and increased GPx levels	[72,73]
		Inhibition of XO	[21]
		Inhibition of NO production and reduction in iNOS	[75,76]
	Anti-inflammatory	Increased Beclin1 expression and obstruction of signaling in the PI3K/AKT/mTOR pathway	[78]
		Inhibition of NF-κB activation and decrease in the level of IL-1β	[79]
		Inhibition of the production of PGE2 and pro-inflammatory cytokines (IL-1β, IL-6, TNF-α, and COX-2)	[80]
	Anticancer	Decrease in tumor volume and tumor burden by increasing p21 CKDIs	[81,82,83]
	Antiemetic	Inhibition of 5-HT3 receptors in the central and peripheral nervous systems	[84,85,86]
	Antimicrobial	Inhibition of bacterial protein synthesis, DNA replication and cell division.	[94]
	Cardiovascular disease prevention	Reducing effect of PCSK9 through a modulatory mechanism of lipid levels	[88]
	Neurodegenerative disease prevention	Inhibition of TNF-α, NO, COX-2, and iNOS	[89]
		Inhibition of glial cell activation	[90]
		Protective effect against amyloid-β	[91]

AKT: protein kinase B; ERS: endoplasmic reticulum stress; MDA: malondialdehyde; GPx: glutathione peroxidase; XO: xanthine oxidase; NO: nitric oxide; iNOS: inducible nitric oxide synthase; PI3K/AKT/mTOR: phosphoinositide 3 kinase (PI3K)/AKT/mammalian target of rapamycin (mTOR); NF-κB: nuclear factor κB; 5-HT3: 5-hydroxytryptamine type3; PGE2: prostaglandin E2; IL-1β: interleukin-1 beta; IL-6: interleukin-6; PCSK9: proprotein convertase subtilisin/kexin type 9; TNF-α: tumor necrosis factor alpha; COX-2: cyclooxygenase-2; CKDIS: cyclin-dependent kinase inhibitors.

### 3.2. Antioxidant Activity of Turmeric

Antioxidants inhibit the formation of FRs and their propagation; they prevent the formation of peroxides and interfere in the FR chain reaction [96]. Antioxidants counteract OS and promote healthy aging. The organism presents endogenous enzymatic antioxidant defenses: GPx, CAT, and SOD and non-enzymatic antioxidant defenses, such as vitamin C, vitamin E, β-carotene, zinc, selenium, and uric acid [97,98].

Curcumin is an effective antioxidant that reduces OS levels. It can act as a chelator of pro-oxidant metals and regulate the activity of numerous enzymes [99]. It can reduce ROS generation, scavenge FRs, and act as a strong inhibitor of advanced glycation end products and lipid peroxidation [100]. There are several mechanisms by which curcumin prevents OS. Curcumin acts as an activator of sirtuin (SIRT)1 and SIRT3, but as an inhibitor of SIRT2. SIRT1 and SIRT3 inhibit OS in cells, whereas SIRT 2 triggers it [100]. SIRT1 prevents the expression and production of iNOS and ROS. This action occurs through p65, leading to the suppression of the NF-ĸB signaling pathway [101]. SIRT 1 is primarily located in the nucleus and activates FOXO 3a protein transcription factors (Forkhead box O) that regulate antioxidant genes (CAT and SOD) by reducing FR levels [102]. Mitochondrial SIRT3 deacetylates and activates several enzymes that are critical in maintaining cellular ROS levels. One of these enzymes is SOD2, which is an important antioxidant enzyme located in the mitochondria. SIRT3 deacetylates two important lysine residues in SOD2, which enhances its catalytic activity. It has also been observed that the catalytic activity of SOD2 decreases when SIRT3 is suppressed [103].

Turmeric/curcumin supplementation might be used as a viable intervention for improving the inflammatory/oxidative status of individuals. In addition, turmeric/curcumin supplementation significantly improved antioxidant activity through reducing MDA levels and SOD activity and enhancing TAC [104]. Panahi et al. [105] observed that the supplementation with a curcuminoid–piperine combination significantly reduced MDA and CRP concentrations and improved serum SOD activities compared with placebo. Short-term supplementation with a curcuminoid–piperine combination significantly improves oxidative and inflammatory status in patients with metabolic syndrome.

Curcumin stimulates peroxisomal enzymes with high antioxidant activity, such as CAT and SOD. These enzymes reduce OS through the peroxisome proliferator-activated receptor alpha/gamma (PPAR) pathway [27]. The reduction of reactive nitrogen and oxygen species is essential for maintaining normal neuronal functions.

Neurodegenerative diseases are associated with increased OS and inflammatory processes. This increased OS is associated with mitochondrial dysfunction, calcium overload, and excitotoxicity.

AD is a neurodegenerative disorder of the elderly. As the prevalence of AD rises in the 21st century, there is an urgent need for the development of effective pharmacotherapies. Physical exercise and healthy diets have been reported to have implications to delay disease progression of AD in elderly individuals and improve cognitive functions in subjects with mild cognitive impairment and in early AD patients. Neuroinflammation and OS have been considered as hallmarks of AD, playing crucial roles in neurotoxicity. For this reason, an adequate antioxidant strategy may improve the treatment of neurodegenerative diseases and dementia [106]. Curcumin inhibits the formation and promotes the disaggregation of amyloid-β plaques, attenuates the hyperphosphorylation of tau and enhances its clearance, binds copper, lowers cholesterol, modifies microglial activity, inhibits acetylcholinesterase, mediates the insulin signaling pathway, and is an antioxidant. In conclusion, curcumin has the potential to be more efficacious than current treatments [107].

Curcumin prevents the aggregation of new existing amyloid deposits, promotes disaggregation of amyloid deposits, and reduces their size in patients with AD. It also binds to redox active metals (copper and iron) by preventing the metal induction of Nf-kB, reducing the inflammatory process associated with the disease. Curcumin prevents cognitive decline associated with aging and dementia [108].

Curcumin suppresses the differentiation of Th17 cells, cells that are an important factor in the development of multiple sclerosis. It also causes the degradation of the myelin sheath of neurons in these patients and decreases inflammation, reducing inflammatory cytokines, NF-kB, and JAK-STAT, AP-1, and AP-1 signaling pathways [5].

Any circumstance harmful to health, such as infections and tissue lesions, leads to an inflammatory process, related to numerous pathological and physiological comorbidities, and produced by a series of responses triggered by the immune system. Inflammation is related to an increase in lipid peroxides, products resulting from lipid peroxidation, and an increase in FRs and inflammatory markers, such as cytokines [109]. Curcumin decreases OS and inflammation by acting through the Nrf2-keap1 pathway; it blocks TNF-α-mediated cell signaling and TFN production by suppressing pro-inflammatory pathways associated with chronic diseases [110]. Curcumin regulates NF-κB, MAPK, AP-1, JAK/STAT, and other signaling pathways, and inhibits the production of inflammatory mediators.

OS is closely related to inflammatory processes by activating transcription factors associated with inflammation. Curcumin increases antioxidant activity due to its effect on NADPH oxidase and increasing the activity of antioxidant enzymes (SOD, CAT, and GPx), and is related to the Nrf2-Keap1 pathway [111,112], which could potentially promote cell survival.

Curcumin, as an antioxidant, reduces lipid peroxidation, a risk factor for cardiovascular disease and atherosclerosis. Curcumin also improves the lipid profile by influencing the same mediators as statins. Additionally, curcumin has the ability to prevent endothelial dysfunction and smooth muscle cell proliferation and migration, which may be beneficial in the treatment of atherosclerosis [17]. Moreover, curcumin may exert antidiabetic effects by increasing the gene expression of PPAR-γ, which has a pleiotropic impact on glucose homeostasis and insulin sensitivity and controls gene expression in glucose metabolism and lipids [113].

Curcumin has also been shown to protect against atherosclerosis in mice lacking apolipoprotein E by inhibiting Toll-like receptor 4 expression [114], in addition to its ability to suppress cholesterol accumulation in macrophage foam cells and atherosclerosis [115].

Curcumin has been shown to have anti-proliferative and anti-metastatic properties [116]. These properties may contribute to preventing the onset and development of cancer. Some of the mechanisms by which curcumin may act to prevent cancer include the following: (a) reduction in OS, as it may reduce the production of ROS that damage DNA and other cellular structures, which may prevent the occurrence of genetic mutations that contribute to the development of cancer; (b) anti-inflammatory action by inhibiting the production of inflammatory molecules, such as TNF-α and IL-6, as chronic inflammation has been linked to the development of some types of cancer, and curcumin’s ability to reduce inflammation may help prevent cancer; (c) induction of apoptosis by cell death programming, known as apoptosis, in cancer cells, which may help prevent cancer cell proliferation, as the induction of apoptosis is linked to the action of curcumin on the proteasome; (d) inhibiting angiogenesis by blocking the formation of new blood vessels that nourish tumors, which may prevent tumor formation and growth acting on transcription factors AP-1 and NF-kB; inhibiting COX-2 and 5-lipooxygenase (5-LOX); limiting the expression of IL-8 in pancreatic cancer and head and neck cancer cell lines; and inhibiting angiogenesis mediated by NO and iNOS. Curcumin can also regulate various cell signaling pathways that are involved in cell proliferation, differentiation, and cell survival, which may help prevent cancer [117].

In summary, curcumin may act in several ways to prevent cancer, including reducing OS, inhibiting inflammation, inducing apoptosis, inhibiting angiogenesis, and regulating cell signaling pathways. However, further studies are needed to fully understand the mechanisms of action of curcumin and its efficacy in cancer prevention and treatment [116,117].

Curcumin has antimicrobial activity against a wide range of microorganisms, including bacteria, such as *Escherichia coli*, *Staphylococcus aureus*, *Pseudomonas aeruginosa*, *Streptococcus pyogenes*, and *Helicobacter pylori*. In addition, curcumin can inhibit the formation of bacterial biofilms, which can increase the efficacy of antimicrobial treatments. A possible antibacterial mechanism of action produced by curcumin is the inhibition of cell division protein FtsZ assembly on the Z-ring and the suppression of bacterial cells [118]; in addition, regarding viruses, it has been found to interfere with virus entry into host cells and reduce viral replication. Among the most important effects produced against viruses is the effect of curcumin against human immunodeficiency virus (HIV). Due to effects on the function of some viral proteins, counting viral integrase, protease, and transactivate of transcription protein (Tat), curcumin can inhibit HIV replication and has also demonstrated antifungal activity.

Curcumin treatment of *Candida species* results in cell death. A possible mechanism to explain cell death could be the acidification of the intracellular environment by inhibiting thymidine 1 and proton extrusion causing the inhibition of hyphal development [119].

Table 5 and Figure 1 summarize the properties and mechanisms of action of turmeric.

**Table 5 molecules-28-04024-t005:** Summary of product characteristics of turmeric.

Extract	Function	Mechanism	References
Turmeric	Antioxidant	Chelates of pro-oxidant metals	[99]
		Strongly inhibits advanced glycation end products and lipid peroxidation	[100]
		Reduces lipid peroxidation	
		Acts as an activator of SIRT1 and SIRT3	[100,101,102,103]
		Reduces MDA levels and CRP	[104]
		Improves serum SOD and CAT activities	[100,105]
		Enhances TAC	[105]
	Anti-inflammatory	Inhibits the production of inflammatory mediators	[111,112]
		Inhibits the production of inflammatory molecules, such as TNF-α and IL-6	
		Inhibits COX-2 and 5-LOX	
	Anticancer	Reduces the production of ROS that damage DNA and other cellular structures	[116,117]
		Has anti-inflammatory action	
		Induces apoptosis	
		Inhibits angiogenesis	
	Antimicrobial, antiviral and antifungal activity	Increases the efficacy of antimicrobial treatments	[118,119]
		Inhibitions FtsZ assembly on the Z-ring and the suppression of bacterial cells	
		Inhibits HIV replication	
		Acidifies the intracellular environment	
	Cardiovascular disease prevention	Has ability to prevent endothelial dysfunction and smooth muscle cell proliferation and migration	[114,115]
		Inhibits Toll-like receptor 4 expression	
		Has ability to suppress cholesterol accumulation in macrophage foam cells and atherosclerosis	
	Neurodegenerative disease prevention	Inhibits the formation and promotes the disaggregation of amyloid-β plaques	[106,107,108]
		Attenuates the hyperphosphorylation of tau and enhances its clearance	
		Modifies microglial activity	
		Inhibits acetylcholinesterase	
		Binds to redox active metals (copper and iron)	
		Suppresses the differentiation of Th17 cells	[5]
		Exhibits antioxidant activity	[111,112]

SIRT: sirtuin; CRP: C-reactive protein; MDA: malondialdehyde; SOD: superoxide dismutase; CAT: catalase; TAC: total antioxidant capacity; TNF-α: tumor necrosis factor alpha; IL-6: interleukin-6; COX-2: cyclooxygenase-2; 5-LOX: 5-lipo-oxygenase; HIV: human immunodeficiency virus.

### 3.3. Antioxidant Activity of Cardamom

The natural process of oxidation in living organisms is mediated by the reaction of molecular oxygen with other molecules and leads to the enhancement of FRs and ROS. These ROS and FRs, along with further oxidation products, are associated with an arousal of some degenerative diseases, such as atherosclerosis or cancer, to name a few.

Reports indicate that cardamom (*Elettaria cardamomum*) contains biologically active antioxidant metabolites which render the capacity of scavenging those previously mentioned FRs and provide a defense against oxidation and its damages. The literature describes that it is in seeds and pods where the highest amount of antioxidant molecules is stored. Among the antioxidant components, we can find some phenolic components, such as gallic acid equivalents, with the capacity of avoiding the peroxidation of the linoleic acid system, proving its greater antioxidant activity [120]. Cardamom is a good source of volatile oils, fixed oils, phenolic acids, and sterols [121]. Cardamom’s antioxidant effect is manifested as a reduction in MDA, advanced protein oxidation products (APOP), and nitric oxide (NO) detected in the liver and in plasma [8]. Furthermore, CAT, SOD, and GSH are cellular antioxidants which reduce OS. The activity of CAT was significantly depleted in the cardamom group, as in SOD and GSH.

The most dominant bioactive substances in cardamom are 1,8-cineol (20–60%) and α-terpinyl acetate (20–55%) [34]. Given those components, publications described this extract as an antioxidant and a synergistic agent to favor the skin permeation of certain drugs [122,123], but also as a bacterial inhibitor [124].

OS plays a pivotal role in diabetes complications, both microvascular and cardiovascular. The evolution of diabetes causes mitochondrial superoxide overproduction in vessel endothelial cells, and in the myocardium. Increased levels of intracellular ROS cause defective angiogenesis in response to ischemia, activate a number of pro-inflammatory pathways, and cause long-lasting epigenetic changes which drive persistent expression of pro-inflammatory genes after glycemia is normalized (‘hyperglycemic memory’) [125]. A study in 80 pre-diabetic subjects, who randomly took the cardamom supplement (*n* = 40) or placebo (*n* = 40) during 8 weeks, described that cardamom improved parameters of inflammation and OS (decreasing serum CRP (*p* = 0.02), hs-CRP:IL-6 ratio (*p* = 0.008), and MDA (*p* = 0.009)) compared with the placebo group [126].

Several publications have proven role of OS in obesity and its comorbid factors [127]: triggering the accumulation of white adipose tissue (WAT) and altering food intake, and causing an increase in preadipocyte proliferation, its differentiation, and size [128]. Obesity per se can also induce systemic OS through multiple biochemical mechanisms, such as superoxide generation from NADPH oxidases (NOX), oxidative phosphorylation, glyceraldehyde auto-oxidation, protein kinase C (PKC) activation, and polyol and hexosamine pathways [129]. A randomized study of eighty-seven obese patients [130] with nonalcoholic fatty liver disease (NAFLD) were allocated in two matching arms of a trial, taking 3 g of cardamom or placebo with meals through six capsules 3-times/day, and cardamom (*n* = 43) and placebo (*n* = 44) for 3 months. Green cardamom significantly increased (*p* < 0.05) antioxidant markers (Sirt1) and decreased inflammatory markers (e.g., TNF-α, IL-6, alanine aminotransferase (ALT), and fatty liver status) [131]. Further results of this group determined that when compared with placebo, GC significantly increased endocrine activation factors of brown fat tissue and the sensitivity index to insulin and decreased glycemic and blood lipidic levels (*p* < 0.05) [132]. Similarly, a parallel, double-blind randomized [9], placebo-controlled study carried out in 83 participants with overweight, obesity, and/or type 2 diabetes, proved that 10 weeks of taking 3 g of green cardamom daily favored an increase in antioxidants, such as SIRT1. Unfortunately, no significant changes in other serum lipidic indices (TC, HDL-c, and LDL-c) were found [133]. Furthermore, other studies have also described the capacity of cardamom to reduce inflammation seen by TNF-α, IL-6, and CRP serum levels, and thus also the expression levels of TNF-α and CRP genes (*p* < 0.001) [134].

There is a key background mechanism in hypertension manifested as an increment in OS and alterations in the total antioxidant capacity, commonly described in other cardiovascular diseases [67,135]. In one study, twenty participants [136] recently diagnosed with stage 1 primary hypertension received 3 g of cardamom in two doses during 12 weeks to link the administration of cardamom with decreased systolic, diastolic, and mean blood pressure, together with an increase in fibrinolytic activity and total antioxidant status (*p* < 0.05).

ROS can damage lipids, nucleic acids, and proteins, thereby altering their functions, leading to many pathologies. There is some evidence in different types of cancer about low antioxidative enzyme levels or higher rates of products of lipid peroxidation, such as MDA, as biomarkers to assess the level of OS. Some of the studies proposed their potential in defining the stage of tumor progression. Preliminary results in animal models have described that cardamom compared against benzo(α)pyrene reduced tumor incidence and multiplicity significantly (*p* < 0.001). Moreover, biochemical assays described an enhancement in the hepatic activities of oxidation preventing hormones (such as glutathione-S-transferases, SOD, GPx, and CAT) in mice models treated with cardamom (*p* < 0.01). Furthermore, the non-enzymatic antioxidant glutathione was increased in the cardamom-treated group significantly (*p* < 0.001), whereas the lipid peroxidation level along with lactate dehydrogenase activity exhibited a significant reduction with cardamom treatment compared to the control (*p* < 0.01). These results suggest that cardamom has the potential to become a pivotal chemo preventive agent against forestomach cancer [137].

In a couple of studies in 142 healthy volunteers (age of 55 years or above) selected for a relatively low baseline NK cell activity and a history of recurrent coughs, regular consumption of tea with ginger and cardamom enhanced NK cell activity [138].

When the effect of aromatherapy on postoperative nausea after ambulatory surgery was examined in 301 participants, the groups who inhaled a blend of different compounds including cardamom and ginger essential oils showed significant results in the reduction in nausea levels [139].

A recent study with single-blind, randomized, placebo-controlled design used cardamom inhalation for postoperative nausea and vomiting after spinal anesthesia [140]. The intervention had 70 participants who were distributed in placebo and cardamom groups, and regarding nausea, after controlling the initial severity, the declining extent was more noticeable in the intervention group than in the placebo group.

Recent studies have described the capacity of cardamom to suppress bacterial activity, especially its essential oil against methicillin-resistant Staphylococcus aureus (MRSA) [122,141]. However, currently the mechanism of action that cardamom exerts on MRSA biofilm is unclear. Notwithstanding, some studies have deeply assessed cardamom anti-MRSA biofilm activity by destroying the barrier of the biofilm and causing bacterial loss of metabolic activity. Secondly, some studies have avoided MRSA’s adhesion ability and therefore inhibited the formation of extracellular polymers. Finally, there is a suggestion that points out the ability of cardamom to downregulate MRSA biofilm-forming genes [124].

Cardamom is involved in treating AD with the following mechanisms: anti-cholinesterase activity, induction of endogenous antioxidants, such as glutathione and superoxide dismutase, impeding the production of reactive hydroxyl radicals, preventing the formation of Aβ42 deposits, and protecting cells from iron-induced death [142]. Gomaa et al. [143] observed that the effect of 25 mg/kg cardamom supplementation on AD-induced alterations in an animal model could be positive via ameliorating the alterations produced by decreasing OS and neuroinflammation and through the activation of blunted insulin signal transduction in the brain.

Table 6 and Figure 1 summarize the properties and mechanisms of action of cardamom.

**Table 6 molecules-28-04024-t006:** Summary of product characteristics of cardamom.

Extract	Function	Mechanism of Action	References
Cardamom	Antioxidant	Decreased NO, APOP, and MDA	[8,122,123,126]
		Increased activity of CAT, SOD, and GSH	[8,122,123]
		Increased Sirt1	[122,132]
	Anti- inflammatory	Decreased serum CRP	[126]
		Decreased hs-CRP:IL-6 ratio	
		Inhibited the production of inflammatory markers (IL-6, TNF-α, ALT, and fatty liver status)	[131]
	Anticancer *	Enhanced the hepatic activities of oxidation-preventing hormones (such as glutathione-S-transferases, SOD, GPx, and CAT)	[137]
	Antiemetic	Volatile compounds from essential oil were significantly efficient to reduce nausea level	[139,140]
	Antimicrobial	Enhanced NK cell activity and anti-MRSA biofilm activity and avoided MRSA adhesion ability; cardamom may downregulate MRSA biofilm-forming genes	[122,123,124,141]
	Cardiovascular disease prevention	Decreased systolic, diastolic, and mean blood pressure, and increased fibrinolytic activity	[136]
	Neurodegenerative disease prevention	Exhibited anti-cholinesterase activity, induced endogenous antioxidants, such as glutathione and superoxide dismutase, impeded the production of reactive hydroxyl radicals, prevented the formation of Aβ42 deposits, and protected cells from iron-induced death	[142]

*: Mice model, ALT: alanine aminotransferase; Apop: advanced protein oxidation products; CAT: catalase; CRP: C-reactive protein; GSH: glutathione reduced; GPx: glutathione peroxidase; IL-6: interleukin-6; MDA: malondialdehyde; MRSA: methicillin-resistant *Staphilococus Aureus*; NO: nitric oxide; NF-κB: nuclear factor κB; SOD: superoxide dismutase; TNF-α: tumor necrosis factor alpha.

**Figure 1 molecules-28-04024-f001:**
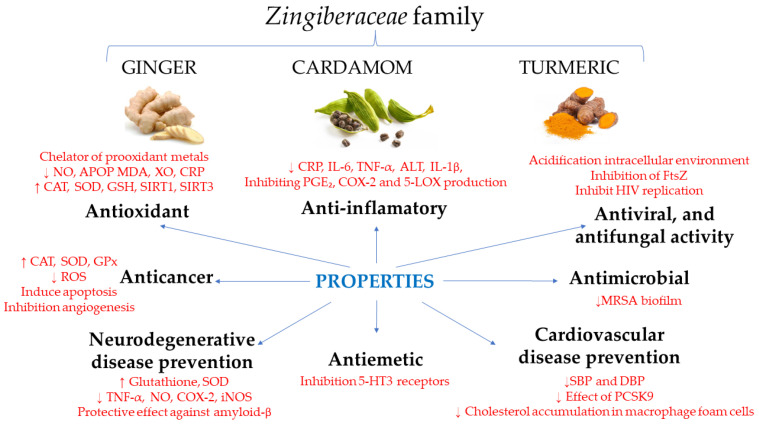
Summary of the functions and mechanisms of action of the 3 products of the *Zingiberaceae* family. Bold color refers to the functions and red color refers to the mechanisms of action.↑: increases; ↓: decreases; NO: nitric oxide; APOP: advanced protein oxidation products; MDA: malondialdehyde; XO: xanthine oxidase; CRP: C-reactive protein; CAT: catalase; SOD: superoxide dismutase; GPx: glutathione peroxidase; ROS: reactive oxygen species; TNF-α: tumor necrosis factor alpha; COX-2: cyclooxygenase-2; iNOS: inducible nitric oxide synthase; 5-HT3: 5-hydroxytryptamine type3; IL-6: interleukin-6; ALT: alanine aminotransferase; IL-1β: interleuikin-1beta; PGE_2_: prostaglandin E_2_; 5-LOX: 5-lypooxygenase; FtsZ: cell division protein FtsZ; MRSA: methicillin-resistant Staphylococcus aureus; SBP: systolic blood pressure; DBP: diastolic blood pressure; PCSK9: proprotein convertase subtilisin/kexin type 9.

## 4. Conclusions and Future Research Lines

Cardamom, turmeric, and ginger are among the most active natural remedies due to their numerous biological properties. They have bioactive compounds with important antioxidant activities. Ginger’s antioxidant capacity is proved by its capacity to decrease AKT-phosphorylation, ERS markers, and MDA and GPx levels, to inhibit XO and NO levels, and to produce PGE2 and pro-inflammatory cytokines (IL-1β, IL-6, TNF-α, and COX-2), as well as via the presence of 5-HT3 receptors in the central and peripheral nervous system. Turmeric improves serum SOD and CAT activity; inhibits the production of inflammatory molecules, such as TNF-α, IL-6, COX-2, and 5-LOX; prevents endothelial dysfunction; and suppresses cholesterol accumulation in macrophage foam cells and atherosclerosis. Cardamom decreases NO, APOP, MDA, CAT, SOD, and GSH activity; induces endogenous antioxidants, such as glutathione and superoxide dismutase; enhances NK cell activity; and inhibits the production of inflammatory markers. Due to all of these properties, cardamom, turmeric, and ginger could be a coadjuvant treatment in different pathologies where oxidative stress plays an important role.

However, the bioavailability of these compounds needs to be optimized, and further research is needed to determine the optimal concentration ranges for these compounds to have effects as antioxidants in the body.

## Data Availability

Not applicable.

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
