# Peer review of "Antioxidant Activity in Extracts from Zingiberaceae Family: Cardamom, Turmeric, and Ginger"

_molecules, 2023, doi:10.3390/molecules28104024_

Round 1
Reviewer 1 Report
- In table 1: Are all the compounds reported or the major ones?
- It is important to place the doses required for the described effect.
- It would be important to mention the mechanism of action for each activity presented for the three primary materials.
- It is important to mention the amount of the components required to have an effect on the organism, or the recommended amount of daily consumption.
-It is important to place the doses necessary to obtain the biactive effect described in each of the studies cited. It would be possible to work on a summary table for each product.
Reviewer 2 Report
The present review evaluates the antioxidant potential of species of the Zingiberaceae family. The topic is relevant but extensively evaluated in the literature, so the novelty should be emphasized by a distinct organization of the information (tables/figures), presentation of other species with antioxidant potential, and perhaps even a comparative evaluation of antioxidant potential, etc. Specific and extensive requirements are needed for the improvement of the information presented.
Shape suggestion
There is no need to abbreviate oxidative stress in the abstract as it is only used once. Abstract and main text should be managed separately in terms of abbreviations. Please check the instructions for authors provided by the journal and revise the whole manuscript in terms of abbreviations.
Abbreviations directly in tables and figures must be explained in the form of legends below them.
Please check the font size for table title 1. The font size must be according to the template provided by the journal. Please revise.
It is not necessary for the heading of the table to contain words written entirely in capital letters.
L177- It is not necessary for " lipid Molecules" to be written in capital letters.
L337- There should be two phrases or reshape the information.
A serious revision at the English level is needed. (e.g., L267- In addition, it has been shown that ginger extract can Inhibiting NF-κB activation and decreasing the level of IL-1β…).
L328- "This action occurs through p65 leading to the suppression of the NF-ĸB signaling pathway [13] (6)." Please clarify the situation in brackets.
L324- bibliographic indexes are separate main text structures and should not be linked to a word. Please revise the whole manuscript.
L422-423-bacterial species should be written in italics.
L441- The name of a plant species is composed of the first word written in capital letters designating the genus, and the second word forming the species is always written in small letters.
L456- "[110,111] [111], but also as a bacterial inhibitor [112]". Please revise and correct.
L473- "hexosamine pathway [117]s". Please correct.
L544- "reactive oxygen species". It should be used in its abbreviated form, ROS, throughout the whole manuscript.
Content suggestions
L11-13 "Life expectancy is increasing, and this is directly related to the increase in diseases." This sentence is confusing, please reshape it because it seems like it's the exact opposite.
The conclusion of the abstract should be reshaped, as it does not conclude the outcomes of this review but only describes what was done in this study. Furthermore, it should also refer to what future research directions this manuscript may refer to.
The concept of traditional medicine and bioactive compounds is much broader, and the first part of the introduction should refer to the broad-general framework, and then address the target Zingiberaceae family. There are many plants with promising antioxidant effects obtained from experimental studies that are worth mentioning in the framework of traditional medicine and the evaluation of antioxidant potential. I suggest checking and referring to: PMID: 35624672, PMID: 35644118, and https://www.researchgate.net/publication/303578834_Comparative_Study_of_Polyphenols_Flavonoids_and_Chlorophylls_in_Equisetum_arvense_L_Populations
The aim of the paper should be improved in the last paragraph of the introduction from the perspective of describing the contribution to the field under analysis and the relevance/novelty of the present manuscript, especially since this is not the first review of this nature.
L83/L111/ A reference is made to several authors, but only one bibliographic reference appears at the end of the mention. Please revise the text or insert more appropriate references.
Section 2 assessing the correlation between oxidative stress and health needs to be developed in terms of the balance of the prooxidant-antioxidant systems and the importance of different biomarkers of oxidative stress. I suggest checking and referring to: PMID: 35883850 and PMID: 35687909. Moreover, reference is made to different neurodegenerative pathologies such as Alzheimer's disease and Parkinson's disease, so information from the literature on the plant-derived family of antioxidants with neuroprotective role should be added. I suggest checking and referring to: PMID: 34030619.
Section 3.2 contains numerous references previously used in the manuscript. It would be advisable to draw on as many new and different bibliographic resources as possible to make the review as comprehensive as possible.
It would be more conclusive to present a figure/table with results on the antioxidant potential evaluation from in vitro studies, animal models, or human subjects, if available. Note, after centralization, the most promising bioactive compounds and the type of study that was conducted.
The conclusions and future research directions section needs to be significantly improved, with the most promising outcomes following the update of the state of knowledge for the evaluated species as well as which future research directions can be opened and in which pathologies they are more likely to be used as adjuvant therapy.
A serious revision at the English level is needed. (e.g., L267- In addition, it has been shown that ginger extract can Inhibiting NF-κB activation and decreasing the level of IL-1β…).
Reviewer 3 Report
The article entitled “Antioxidant activity in extracts from Zingiberaceae family: cardamom, turmeric and ginger” presents a systematic review of the biological activities of different plant extracts and an important advance mainly on their health benefits. However, in order to expand the analysis of the information found, it is important, for example, to make a comparative table with the types of compounds found and the associated activities or mechanisms described. Likewise, it is recommended to make a graph explaining the mechanisms of action of antioxidants. Likewise, in the introduction it is necessary to unify in one paragraph the structure of the document (see lines 124-16 and 131-133).
Round 2
Reviewer 2 Report
The authors have significantly improved the manuscript based on the suggestions received.
Reviewer 3 Report
The recommendations were satisfactorily addressed